# An Evaluation of the Supplementation of Rumen-Protected Lysine and Methionine on the Lactation Performance of Fall Parturition Grazing Holstein Cows in Southern Chile

**DOI:** 10.3390/ani13193118

**Published:** 2023-10-06

**Authors:** Pedro Melendez, Jan Möller, Alejandra Arevalo, Claudio Stevens, Pablo Pinedo

**Affiliations:** 1Jockey Club College of Veterinary Medicine and Life Sciences, City University of Hong Kong, Kowloon, Hong Kong SAR, China; 2Fundo Los Laureles, Chahuilco, Región de Los Lagos, Río Negro 5390000, Chile; jamamoller@gmail.com (J.M.); a.arevalo.vet@gmail.com (A.A.); 3Independent researcher, Valle del Limarí, Santiago 8320000, Chile; cstevens@feedsyr.cl; 4Department of Animal Sciences, Colorado State University, Fort Collins, CO 80523, USA; pablo.pinedo@colostate.edu

**Keywords:** grazing, Holstein, milk, lysine, methionine, urea, fat, protein

## Abstract

**Simple Summary:**

This study assessed the supplementation of two limiting milk production amino acids for dairy cows (methionine and lysine) on the lactational performance of fall-calving grazing dairy cows from Chile. Four groups were compared and 26 cows per group were randomly assigned to the following groups: (i) a control group without supplementation; (ii) a lysine group; (iii) a methionine group; and (iv) a lysine plus methionine group. Supplementation was conducted from 2 to 70 days postpartum. The group supplemented with both amino acids produced more milk protein and fat than the control group. In conclusion, the supplementation of methionine and lysine improved milk protein and fat yield in grazing dairy cows with fall parturitions; therefore, they can be used as a strategy to improve the performance of lactation in cows.

**Abstract:**

The aim of this study was to evaluate the effect of the supplementation of rumen-protected (RP) methionine and lysine on milk yield, solids, and body weight over time on fall-calving grazing multiparous Holstein cows from Chile. Four treatment groups were studied and compared for the outcomes over time. The treatments were as follows: (i) CON: control (n = 26); (ii) RP lysine group (LYS; 20 g per cow per day; n = 26); (iii) RP methionine group (MET; 20 g per cow per day; n = 26); and (iv) LYS × MET (RP lysine and RP methionine 20 g of each amino acid per cow per day; n = 26). Data were analyzed with general linear mixed model ANOVAs for repeated measures to primarily test the main effects of each amino acid and their interactions. The supplementation of the amino acids was conducted from 2 to 70 days postpartum. Overall, milk production tended to be higher in the MET and in the LYS × MET group when compared to the control group. Furthermore, CON produced significantly less milk protein (kg) and milk fat (kg) than the LYS, MET, and LYS × MET groups. Milk urea tended to be lower during the entire study in the CON group than the rest of the groups. There was a trend for a reduction in the losses of postpartum body weight in the LYS × MET than the CON. In conclusion, RP methionine and lysine improved milk fat and protein yield in grazing multiparous cows with fall parturitions; consequently, both RP amino acids can be used as a strategy for improving grazing cows’ production performance.

## 1. Introduction

The “Cornell Net Carbohydrate and Protein System” (CNCPS) model has the ability to more accurately predict metabolizable energy (ME) and metabolizable protein (MP) to synthesize milk in dairy cows. The model offers a better understanding of how dietary protein can be more efficiently used to produce milk and improve fertility without increasing nitrogen emissions [1,2,3,4,5]. In fact, version 6.5 of the model has established the amount of the most limiting amino acids (AA) for milk production (methionine and lysine) to maximize the content of protein in milk. As a result, lysine and methionine at MP levels of 7.00 and 2.60%, respectively, have been recommended to maximize protein yield, and lysine and methionine at MP levels of 6.77 and 2.85%, respectively, have been recommended to maximize the percentage of milk protein [1,5,6,7]. Nevertheless, modern dairy cows barely cover the requirements of lysine and methionine with the available feedstuffs currently used for dairy cows [7]. As a result, the supplementation of rumen-protected (RP) lysine and methionine has shown positive effects on cows’ lactational performance [8,9,10,11]. Nevertheless, dairy cattle managed in pastures also have greater difficulties in meeting their AA requirements due to the typical nutritional restrictions of grazing systems [12,13].

In a recent investigation that was conducted in Holstein cattle with spring parturitions under grazing management in the southern region of Chile, RP lysine and methionine was discovered to have a positive impact on milk production and components throughout the first 80 days in milk (DIM) [14]. The supplementation of lysine had a better impact on primiparous cows, while methionine only improved the lactational performance in multiparous cows. However, the nutritional composition of spring/summer grasses significantly differ from that of fall/winter grasses. Therefore, the impact of lysine and methionine supplementation could be different, depending on whether the cows are delivering either in spring or fall [15]. In fact, the main botanical specie of Chilean pastures is ryegrass (*Lolium perenne*), and the differences in its nutritional composition between fall/winter and spring/summer are remarkable [16]. Notwithstanding, the major difference between both seasons is the amount of dry matter (DM) available for grazing. In spring (September to December), grass availability ranges from 10 to 16 kg of DM per cow per day; however, in fall (March to June), the grass available is reduced to 4 to 6 kg of DM per head per day, mostly due to low environmental temperatures [13,15].

In this scenario, grazing from DM is restricted to cows with fall parturitions (35% of the total herd). As a result, the cow’s diet must be complemented with winter crops (fresh oats), and/or conserved forages (grass silage, grass haylage, grass hay, corn silage, straw) offered as a partial mixed ration (PMR), including concentrates [15]. Accordingly, the balance for methionine and lysine is negative [14]; consequently, providing both AAs as a rumen bypass product may also be beneficial for improving the lactational performance of fall parturition cows, as in spring parturition cows [14]. Our hypothesis was that the supplementation of RP lysine and methionine to reach a lysine-to-methionine ratio of 2.70, expressed as grams per Mcal of ME, would have better productive responses than in groups with different ratios. Therefore, our objective was to evaluate the supplementation of RP lysine and methionine on milk yield, fat (grams and percentage), protein (grams and percentage), urea (mg/L), and body weight (BW) over time in grazing multiparous Holstein cows with fall parturitions in Southern Chile.

## 2. Materials and Methods

### 2.1. Dairy Herd

This study was conducted under the standards of the Animal Well-Being convention of the Chilean Dairy Consortium and with the consent of the owner of the farm (Mr. Ricardo Kramer, Los Laureles Dairy Farm, Osorno, Chile). This investigation complements previous research completed with cows under spring calving management [14]. The study farm was situated at longitude −73.27 and latitude −40.70 and at an altitude of 109 m. Lactating cows were managed under grazing conditions. Weather parameters [average (range)] in the area were as follows: temperature = 11.4 °C (7.7 to 15.6); monthly precipitation = 112 mm (44 to 206); and accumulated yearly precipitation = 1325 mm [17].

The operation included 1232 Holstein lactating cows with a 305 days’ milk production of 7580 kg. Parturitions were concentrated in two periods [35% in the fall season (from March to May) and 65% in the spring season (from July to September)].

Lactating cows were handled according to their parity number and DIM in different groups. The pasture consisted of perennial ryegrass (*Lolium perenne*). The cows were fed based on the requirement system proposed by the CNCPS model, version 6.55 [5] (Table 1). The amount of fresh grass offered to the cows depended on the season of the year due to the fact that the growth of the plant depended on the environmental temperature of the soil, affecting the availability of forage in the coldest months of the year (May to August). Part of the concentrate (70%) and conserved forage (hay and silage) were fed as a PMR before each milking during the entire fall/winter season. The rest of the concentrate (30%) was fed in individual milking parlor feeders twice a day during each milking. The amount of concentrate fed in the individual feeders was based on the DIM and lactation number.

At 45 to 75 days before expected parturition (BEP), the cows were dried off and set aside to pasture until 3 weeks BEP, where they remained housed together until parturition. During the prepartum period, the cows received an anionic PMR to prevent clinical hypocalcemia, targeting a dietary cation anion difference (DCAD) of −109 mEq/kg DM using the formula [Na] + [K] – [Cl] + [S] [18]. The anionic product contained ammonium chloride and hydrochloric acid (Meganion, Origination O2D Inc., Maplewood, MN, USA).

The calving took place in an adjacent barn to the parlor. Parturition monitoring was conducted every 4 h during the whole calving season. Offspring born from normal deliveries were immediately put in individual hutches. After calving, the dam was directed to a postpartum group until 30 DIM. The health status of this group was monitored every other day for the first two weeks postpartum after the morning (AM) milking. The parameters evaluated were the presence of retained fetal membranes, puerperal metritis, body temperature, clinical mastitis, subclinical ketosis using a hand-held electronic meter, and the presence of a displacement of the abomasum. Treatments were provided according to the criteria of the herd veterinarian.

If the cows were clinically normal at 30 DIM, they were transferred to a group of high milk production until they became pregnant or their milk production level dropped below the dairy cow average; they were then moved to a low-producing group. The cows remained in this group until dry-off. The cows were weighed daily on an electronic scale as they walked through an exit line, following the morning milking. Milk weights and BW were stored in the main farm computer processor using a commercial software (Afimilk, Kibbutz Afikim 1514800, Israel). The voluntary waiting period for breeding purposes was 50 days. The pre-synchronization and synchronization of an ovulation program and timed artificial insemination (TAI) at 16 to 20 h post GnRH dose was carried out as a routine [19,20]. Cows diagnosed open at 32 days after insemination were resynchronized for a second TAI. After the second insemination, the cows were handled in pastures with bulls for a natural service program until the completion of a period of 90 days.

### 2.2. Experimental Design

This investigation tested the significance of RP methionine and/or lysine supplementation to reach different lysine-to-methionine ratios expressed as grams of the AA per Mcal of ME. Due to the limited number of heifers delivering during the fall season as compared to the spring season, this study only considered multiparous cows. To identify a difference in the milk yield of 2.0 ± 2.5 kg between treatment groups and a control group, taking a 95% of confidence (Type I error 5%) and an 80% power of the test (Type II error 20%) into consideration, a number of 26 cows per group was considered a sample size [21,22].

Cow assignment was conducted between March and June 2020 (fall season) using a stratified sampling procedure based on the lactation number. Only multiparous cows were subjected to each group, constituting a proportional sample of 46% of the cows of parity 2 lactations (12 cows per group) and 54% of the cows of parity 3 or more lactations (14 cows per group). Four experimental groups were planned in a 2 × 2 factorial design to test the interaction of the main treatment effects (lysine and methionine) and the interaction of the main effects with time, building up the general linear mixed model ANOVA for repeated measures. The experimental groups were as follows: (i) control (CON), no supplementation; (ii) lysine (LYS), 20 g per cow per day of RP lysine that was supplemented during morning milking on parlor individual feeders; (iii) methionine (MET), 20 g per cow per day of RP methionine that was supplemented the same as LYS; and (iv) lysine and methionine (LYS + MET), 20 g per cow per day of RP lysine and methionine, respectively. Both AAs were commercial products (Kemin Industries Inc., Des Moines, IA, USA), and the quantities to be supplemented were calculated to reach different lysine-to-methionine ratios in grams per Mcal of ME, as recommended by the CNCPS model, version 6.5 [5]. The supplementation of the AAs started at day 2 postpartum until 70 DIM. These were only offered during the morning milking in the parlor individual feeders.

The concentrate was fed to the cows daily in a specific amount according to the lactation number of the cow. An amount of 30% of the concentrate was provided in the parlor feeder, and the remaining 70% was fed in a PMR offered in a walk-through barn before each milking. The PMR was fed to the cows during the entire fall/winter season. The cows took no more than 1 h to consume the PMR. All feed (forages and concentrates) were analyzed for nutritional composition with wet chemistry every other week (CVAS, Waynesboro, PA 17268, USA). The diets and forage composition are described in Table 1 and Table 2. The wet chemistry AA composition of the feedstuffs is shown in Table 3.

The experimental cows from the four groups were maintained and handled together in the same lot, grazing daily in the same pasture and receiving the same PMR before each milking. After each milking, the cows returned to the pasture. Overall, the cows remained in pasture approximately 18 to 20 h a day, because both milking procedures and PMR consumption took between 4 and 6 h per day. Since the growth of the pasture was variable every day, the amount of grass offered daily was measured with an electronic Rising Plate Meter with a diameter of 36 cm and weight of 315 g (FARMWORKS Ltd.^®^, Feilding, New Zealand), in accordance with Cárdenas et al. [23].

Daily milk production and BW were the outcome variables of this study. Due to computer modelling restrictions, both variables were averaged weekly for a period of 10 weeks. Other outcome variables were milk fat and protein (kg/day and percentage) and milk urea concentrations (mg/L). For milk fat and protein, a composite milk sample from the four quarters was collected weekly during morning milking. The samples were transported immediately after collection to an official accredited laboratory (Cooprinsem, Osorno, Chile). The samples were analyzed with a MilkoscanTM FT1 (FOSS, Nils Foss Allé 1, DK-3400 Hilleroed, Denmark).

### 2.3. Diet Formulation

Due to the fact that the growth rate of the pasture is low during the fall/winter seasons, the total amount of forage for grazing offered per cow per day was not greater than 5 kg of DM. The daily supply of grass was managed through an electric fence system. The complement of the total diet was offered as a PMR and concentrate in the milking parlor.

The diets were balanced in consideration of the average cow with a parity of 3, at 630 kg of BW at parturition, producing 35 kg of milk (4.1% fat, 3.6% protein) at 75 DIM, calving with a BCS of 3.5 (scale 1 to 5), and losing no more than 0.75 units of BCS until 75 DIM. Diet formulation was carried out with a commercial software (NDS Professional, Rum&n Sas, 42123 Reggio Emilia–Italy), which uses the CNCPS v.6.55 model as a nutritional computational base [5]. After parturition, the cows were moved from the maternity barn to the postpartum group. All of the treatment cows were handled together, grazing at the same pasture. Only the amount of concentrate offered in the individual parlor feeders was different according to the lactation number of the cow, though the groups were stratified and balanced by parity number. On average, the control diet had a negative balance of methionine (−8.2 g) and lysine (−4.2 g) (Table 4). Accordingly, the experimental diets contained 20 g of RP methionine and 20 g of RP lysine. In this way, the average AA daily balance increased to +5.2 g for methionine and +4.5 g for lysine.

### 2.4. Statistical Analysis

All independent variables (milk production, milk fat, milk protein, milk urea, and BW) were investigated, building up general linear mixed models ANOVA for repeated measures. For this purpose, the PROC MIXED program of SAS 9.4 was utilized (SAS/STAT, SAS Inst. Inc., Cary, NC, USA) [21,22].

The variables were as follows: the main effect of the treatment (LYS and MET), the main effect of time (day), the effect of the interaction Lys × Met, and the effect of the interaction of the main effects with time. The effects of the BCS at calving and lactation numbers were included as covariates in the model. The cow was nested within her respective treatment group as a random variable.

Models were defined as:y_ijklm_ = µ + Lys_i_ + Met_j_ + (Lys × Met)_ij_ + Time_k_ + (Lys × Time)_ik_ + (Met × Time)_jk_ + (Lys × Met × Time)_ijk_ + BCS_l_ + Par_m_ + e_ijklm_
where:

y_ijklm_ = milk yield, body weight, milk components

µ = population mean

Lys_i_ = the main effect of lysine

Met_j_ = the main effect of methionine

(Lys × Met)_ij_ = the interaction effect of lysine with methionine

Time_k_ = the effect of time (days postpartum)

(Lys × Time)_ik_ = the interaction effect of lysine with time

(Met × Time)_jk_ = the interaction effect of methionine with time

(Lys × Met × Time)_ijk_ = the effect of the interaction of lysine with methionine with time

BCS_l_ = the effect of the body condition score at calving

Par_m_ = the effect of parity

e_ijklm_ = error term

The level of significance was considered when *p* ≤ 0.05, and a trend was established when 0.05 < *p* ≤ 0.15.

## 3. Results

### 3.1. Milk Production

The main effect of the treatments, the interaction of the treatments, the day, and the interaction of the treatments by day on milk yield are shown in Table 5. The main effect of methionine was not significant. The Met group tended to produce more milk during the entire period than the Con and Lys groups (*p* = 0.10). There was also a trend for the interaction of Lys × Met to produce more milk than the Lys and Con groups. There was no interaction of Lys × day, Met × day, and Lys × Met × day on milk yield. The day was also a significant effect in the model (*p* < 0.001), meaning that all groups peaked in milk yield between 21 and 49 DIM (Figure 1).

### 3.2. Milk Fat

For the milk fat yield, there was a significant positive response for the main effect of lysine and methionine and the interaction effects of both AAs. Only the interaction of Lys × Met × day was significant, meaning there was a higher milk fat yield curve over time when compared to the control group (Table 5, Figure 2). For the milk fat percentage, only the interaction effect of Lys × Met and the effect of the day were significant variables in the models (Table 5, Figure 3). Lys × Met had a higher milk fat percentage than the Lys and Met groups but was not different than the control group.

### 3.3. Milk Protein

For the milk protein yield, there were positive responses for the main effect of methionine and the interaction of Met × Lys, Met × day, and Met × Lys × day as well as a trend for higher protein yield for the Lys group (Table 5, Figure 4). However, for the milk protein percentage, only the main effect of lysine and the interaction of Lys × day and the day effect were significant variables in the model. There was a trend for a higher protein percentage in favor of the interaction of Lys × Met (Table 5, Figure 5).

### 3.4. Milk Urea

In the model for milk urea, there were neither a main treatment effect nor interaction effects (Table 5, Figure 6); however, there was a trend for the interaction of Lys × day, Met × day, and Lys × Met × day for a higher concentration of milk urea over time than in the control group.

### 3.5. Body Weight

For BW (kg), there was a trend for the interaction effect of Met × Lys to lose less BW during the postpartum period than the rest of the group.

## 4. Discussion

This study is complementary to previous research carried out in Chile in cows with spring parturitions [14]. This research was conducted on the same farm, which is a commercial grazing dairy farm from the southern part of Chile, which allowed for the same management conditions and restrictions. Unfortunately, in this investigation, it was not feasible to evaluate primiparous cows, because the calving structure of Southern Chilean dairy farms mostly plans for heifer parturitions in the spring season. Therefore, one of the differences between both studies is that during spring parturitions, it was possible to compare primiparous and multiparous cows, unlike this investigation in which the evaluation was only conducted on multiparous cows.

A grazing Holstein cow weighing 630 kg, being 75 days in milk, and producing 35 kg/d (4.1% fat, 3.6% protein) under the environmental conditions of the Southern Hemisphere has a requirement of 62.73 Mcal of ME and 2,783.6 g of MP. Assuming an availability of ryegrass at a DM basis of 5 kg for grazing, 15 kg of a PMR (5 kg of corn silage, 4 kg of ryegrass silage, and 6 kg of concentrate), and 4 kg of concentrate offered in the milking parlor, there is a limitation of methionine and lysine, reaching only 91.0% and 90.0% of their requirements, respectively [5,7]. As a result, the control group of this investigation had a shortage in both AAs at the beginning of this study.

One of the major drawbacks of nutritional trials conducted on grazing cattle is the evaluation of individual DM intake. In grazing dairies, it is difficult and challenging to properly quantify individual feed intake. This was not the exception for this investigation; therefore, we were not able to measure and report individual DM intake. That AA supplementation may affect DM intake and that the potential differences in milk or fat and protein yield are more the reflection of a greater or lesser consumption of DM than the intrinsic effect of the supplemented AA are considerations that are not ruled out. Although this major weakness is thought-provoking, a better approach to obtain an estimation of the DM intake would be to compare treatment groups and obtain an average intake per group by subtracting the amount of DM at the exit of the pasture to the amount of DM at the entrance of the pasture. This methodology requires having several groups per treatment since the experimental unit is the group and not the animal anymore. Unfortunately, this kind of experimental design is unfeasible under commercial conditions.

The other major drawback of this study lies on the high variability of the nutritional composition of the grass throughout the time of the investigation. Table 2 summarizes the nutritional composition of the fresh pasture throughout the investigation. Dry matter ranged from 12.8% to 16.6%, aNDFom ranged from 38.4% to 46.3%, crude protein ranged from 23.0% to 30.8%, and soluble sugars ranged from 3.8% to 12.2%. These values are consistent with other nutritional evaluations reported in Chile for ryegrass when comparing differences between fall/winter and spring/summer, where DM (%) was 15.73 ± 4.43 vs. 18.45 ± 4.48, respectively. For crude protein content (% DM), it was 25.21 ± 4.31 vs. 22.50 ± 3.90, respectively, and for neutral detergent fiber content (% DM), it was 38.60 ± 4.52 vs. 40.73 ± 4.31, respectively [16]. Undoubtedly, this enormous variability, including the AA composition variability of feedstuffs, may impact the results and conclusions of this investigation. Nevertheless, this investigation was designed to be able to extrapolate the results to commercial dairies that manage the herd in a similar way to the method used in this investigation.

One of the advantages of using this dairy operation was that the farm has a very efficient computerized record system that allowed for the evaluation of daily milk yield and BW of the cows. Moreover, the facilities and management of the dairy permitted us to individually supplement both RP AAs and allowed us to keep the experimental animals in the same group, which helped to enormously reduce the sources of variation.

In general, our study shows differences and trends in favor of the treated groups when compared to control animals. Although there was a trend for a higher milk production in the Met group, the impact of methionine and lysine on fat and protein yield was more markedly and consistently higher than the Con group. Overall, the supplementation of both RP AAs over time seemed to be more effective than the supplementation of each AA separately to favorably modulate the fat and protein yield, but the percentages of solids were moderately affected by the treatments, because the increase in milk production could have masked the increase in the percentage of solids.

The control group had a tendency for lower milk urea content during the entire study period when compared to treated groups. However, all of the groups had similar urea concentrations of approximately 300 mg/L, which are within the normal ranges for dairy cows. Taking into account that there was only a tendency towards a lower urea concentration in the control group than in the rest of the groups and that there are many other factors that may influence the final concentration of urea in milk, it is not reasonable to imply that these trends are in fact due to a higher N load by the RP AAs. In fact, as Table 2 shows, perennial ryegrass during the study period had a large variable content of crude protein (23 to 31%) and soluble protein (7 to 11%) that could also impact nitrogen efficiency. Dairy cows consuming high quality perennial ryegrass may lead protozoal contributions to the small intestine, as a microbial supply, to reach up to 23%, which may represent the upper limit of the protozoal contributions (22–25%) of the total microbial yield, increasing the total nitrogen supply at the duodenal level [24]. Therefore, Cornell researchers are still refining their model, intending to develop better integration of nitrogen efficiency, which will highlight the innate capacity of ruminants to recycle nitrogen, whereas excessive nitrogen losses in feces will be reduced without affecting productive responses [7,25,26].

As with the study conducted during the spring season [14], in this investigation, we planned to work with the same recommended amounts of 3.16 and 1.17 g of lysine and methionine per Mcal of ME, respectively, to maintain the optimal ratio of 2.70 recommended by the CNCPS model [5]. Only in the LYS × MET group, the lysine-to-methionine ratio was close to the recommended value of 2.70 (ratio = 2.68, Table 4), although the amount of both AAs in grams per Mcal of ME only reached 2.81 and 1.05 g, respectively. The results of this investigation show consistent outcomes in favor of the LYS × MET group when compared to the trial of the spring season in which only the LYS group saw positive effects in primiparous cows, and the MET group did the same only in multiparous cows [14]. As a result, this investigation reports similar results to studies carried out in animals managed under complete confinement, where the supplementation of both AAs could reach a ratio of 2.70, which has shown positive effects on milk production and/or its protein and fat content [10,27,28]. It is suggested that the supplementation of both AAs during the fall season benefits the levels of fat and protein for the reason that pasture consumption is limited to only 5 kg of DM per cow daily, receiving a higher proportion of a PMR, which is greatly resemblant to a conventional confined herd fed a 100% TMR. Other publications have reported no effect of both RP AAs on milk production and solids [29]. Finally, we do not rule out that the balance of diet AAs other than lysine and methionine could mask the effect of the tested AAs, which could impact the productive responses at different levels [7,30].

## 5. Conclusions

The supplementation of RP methionine and lysine in different combinations tended to improve the milk yield, and significantly increased fat and protein milk content in grazing cows with fall parturition when compared to a control group without supplementation.

The impact of AA supplementation seems to be dependent on the amount of the daily DM of pasture offered to cows, due to the divergent results reported in this investigation and a previous study conducted during the spring season. Further research is encouraged on grazing dairy cattle with different season parturition in order to complement the results of this line of investigation; however, these results suggest that the supplementation of RP lysine and methionine is a valid strategy to improve cow lactation performance under grazing conditions.

## Figures and Tables

**Figure 1 animals-13-03118-f001:**
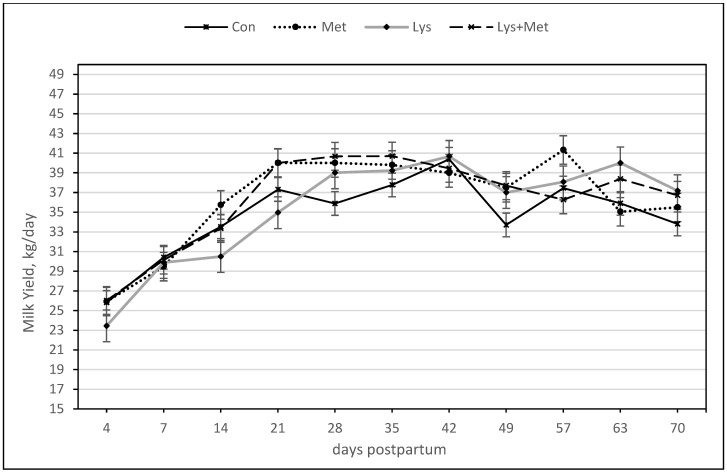
Milk yield (kg/day) during the first 70 days postpartum in fall grazing Holstein cows. The Met group (*p* = 0.10) and Lys+Met group (*p* = 0.14) tended to produce more milk than the Con and Lys groups during the entire period.

**Figure 2 animals-13-03118-f002:**
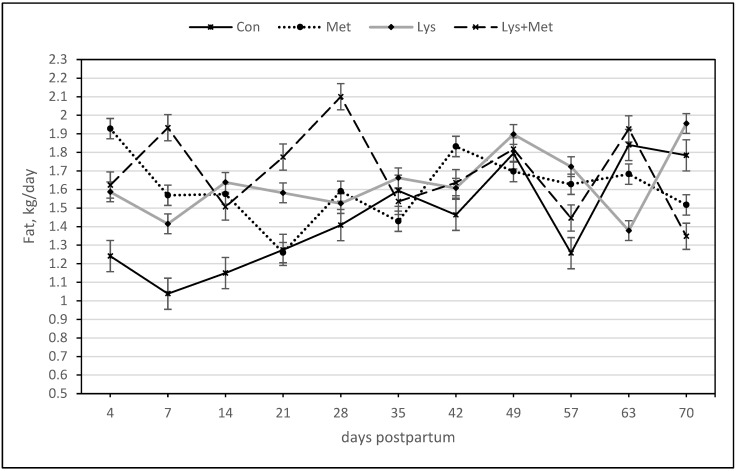
Milk fat (kg/day) during the first 70 days postpartum in fall grazing Holstein cows. The significant main effect of lysine and methionine and the interaction effect of both amino acids (*p* ≤ 0.05). The interaction of Lys × Met × day was also significant, meaning there was a higher milk fat yield over time when compared to the control group.

**Figure 3 animals-13-03118-f003:**
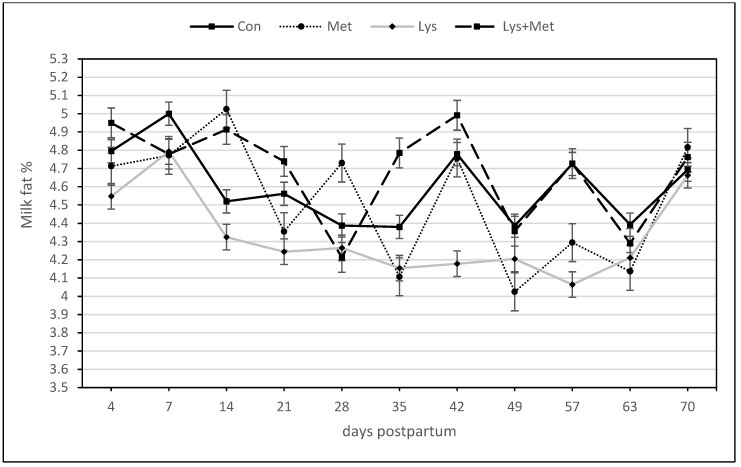
Milk fat percentage during the first 70 days postpartum in fall grazing Holstein cows. A significant interaction effect of lysine and methionine (*p* ≤ 0.05).

**Figure 4 animals-13-03118-f004:**
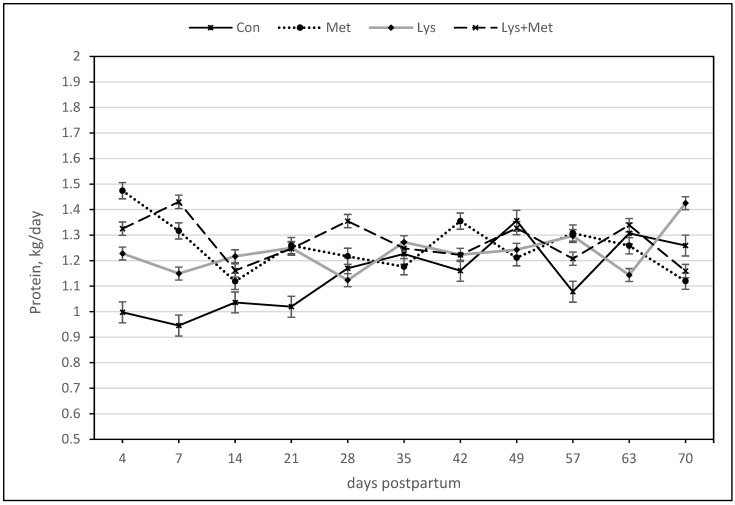
Milk protein (kg/day) during the first 70 days postpartum in fall grazing Holstein cows. The main effect of methionine and the interaction of Met × Lys, Met × day, and Met × Lys × day (*p* ≤ 0.05).

**Figure 5 animals-13-03118-f005:**
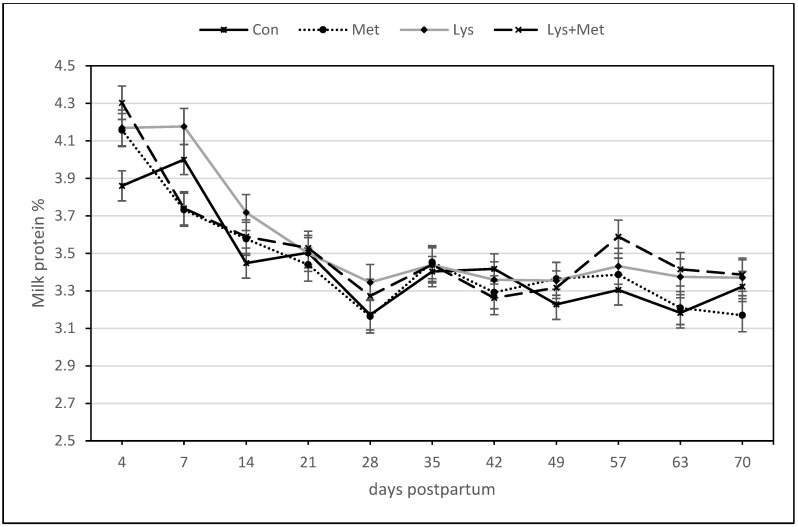
Milk protein percentage during the first 70 days postpartum in fall grazing Holstein cows. The significant main effect of Lys and the interaction of Lys × day (*p* ≤ 0.05).

**Figure 6 animals-13-03118-f006:**
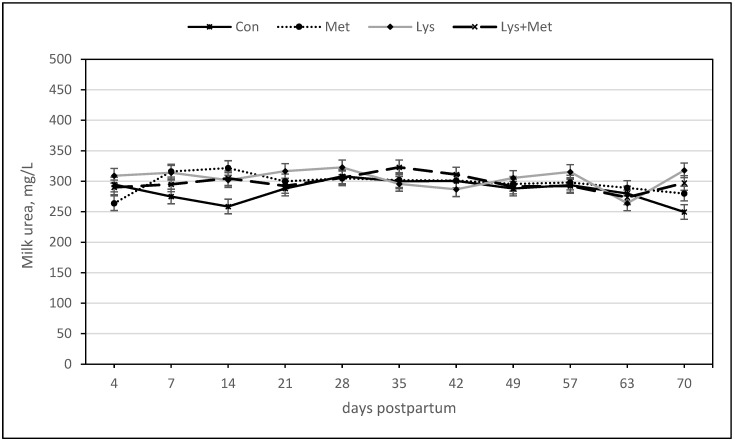
Urea (mg/L) during the first 70 days postpartum in fall grazing Holstein cows.

**Table 1 animals-13-03118-t001:** The constituents of the diet (kg, dry matter [DM]), average nutrient composition (6 samples wet chemistry), and DM intake predicted for an average cow (630 kg BW, BCS 3.0) producing 35 kg of milk (4.0% fat, 3.6% protein) at 75 days in milk.

Ingredient	Kg DM
Concentrate ^1^	8.77
Molasses	1.05
Beet pulp	0.44
Ryegrass silage	4.0
Corn silage	5.1
Pasture (ryegrass)	5.0
Barley straw	0.5
Total kg	24.86
**Nutrient composition**	**% DM**
Dry matter (%)	33.46
Crude protein	16.97
aNDFom ^2^	30.30
Starch	25.51
Sugars	4.40
Fat (EE)	3.32
Ash	6.69
Ca	0.75
P	0.39
Mg	0.28
K	1.94
S	0.25
Na	0.15
Cl	0.43
**Amino Acids**	**% DM**
Met	0.25
Lys	0.81
Arg	0.88
Thr	0.65
Leu	1.32
Ile	0.69
Val	0.83
His	0.39
Phe	0.81
Trp	0.24

^1^ Concentrate supplemented in the parlor. ^2^ Neutral detergent fiber starch- and ash-free.

**Table 2 animals-13-03118-t002:** The wet chemistry nutritional composition (average and range of 6 samples taken every other week) of forages at a DM% basis.

	Fresh Pastures (*Lolium perenne*)	Pasture Silage (*Lolium perenne*)	Corn Silage	Barley Straw
Dry matter (%)	14.3 (12.8–16.6)	46.1 (39.4–53.4)	33.1 (31.5–34.8)	91.2 (90.3–92.0)
Crude protein (%)	27.2 (23.0–30.8)	14.2 (10.1–18.7)	6.0 (5.8–7.2)	3.8 (3.0–4.9)
Soluble protein (%)	9.0 (6.7–10.6)	7.3 (3.9–10.9)	3.2 (2.8–4.0)	1.58 (1.35–1.98)
aNDFom ^1^ (%)	41.6 (38.4–46.3)	46.5 (41.3–51.1)	49.2 (43.4–52.3)	76.3 (68.8–80.5)
Sugars (%)	8.6 (3.8–12.2)	12.5 (11.3–13.7)	3.8 (3.2–4.5)	3.52 (3.23–3.89)
Starch (%)	1.1 (0.2–1.6)	1.0 (0.9–1.2)	22.6 (21.0–25.0)	1.3 (1.1–1.5)
Soluble fiber (%)	6.0 (1.5–11.3)	9.2 (7.1–11.1)	4.4 (3.8–5.0)	5.91 (4.3–7.1)
Fat (%)	5.0 (3.8–6.0)	3.8 (3.1–4.6)	2.7 (2.1–3.5)	1.75 (1.1–2.5)
Ash (%)	10.6 (9.2–11.4)	9.6 (7.9–11.3)	4.7 (4.3–5.5)	7.5 (5.5–8.9)

^1^ Neutral detergent fiber starch- and ash-free.

**Table 3 animals-13-03118-t003:** The wet chemistry amino acid composition (average of 2 samples) of ingredients and total diet DM% basis.

	MET	LYS	ARG	THR	LEU	ILE	VAL	HIS	PHE	TRP
Pasture (April 2020)	0.30	0.87	0.74	0.74	1.33	0.71	0.94	0.35	0.86	0.38
Mix milking parlor ^1^	0.21	0.52	0.70	0.45	1.23	0.46	0.56	0.33	0.60	0.12
Corn silage	0.10	0.18	0.15	0.22	0.55	0.22	0.29	0.11	0.25	0.05
Ryegrass silage	0.25	0.69	0.64	0.70	1.29	0.75	1.03	0.35	0.92	0.22
Concentrate PMR ^2^	0.68	3.04	3.59	1.95	3.76	2.24	2.35	1.30	2.48	0.68
Molasses	0.02	0.09	0.42	0.13	0.31	0.38	0.29	0.14	0.23	0.04
Barley straw	0.07	0.13	0.08	0.16	0.25	0.21	0.16	0.07	0.18	0.05
Beep pulp	0.17	0.54	0.68	0.43	0.72	0.40	0.60	0.34	0.45	0.18
Total diet	0.25	0.81	0.88	0.65	1.32	0.69	0.83	0.39	0.81	0.24

^1^ Concentrate fed in an individual milking parlor feeder twice a day during each milking; ^2^ Concentrate fed as part of a partial mixed ration offered twice a day in a walk-through feeding barn before each milking.

**Table 4 animals-13-03118-t004:** The amount of lysine and methionine supplemented and grams per Mcal of ME of both AAs reached in the diets of each experimental group (CNCPS v.6.55 model ^a^) for a cow of 630 kg in BW, a BCS of 3.0, at 75 DIM, and producing 35 kg of milk. Rumen protected = RP; Metabolizable Energy = ME.

CONTROL (CON) (n = 26)	Supplied	Optimal
RP Methionine (g)	0	-
RP Lysine (g)	0	-
Methionine g/Mcal ME	0.85	1.17
Lysine g/Mcal ME	2.64	3.16
Lysine-to-Methionine ratio	3.10	2.70
**LYSINE plus METHIONINE (LYS + MET) (n = 26)**		
RP Methionine (g)	20	-
RP Lysine (g)	20	-
Methionine g/Mcal ME	1.05	1.17
Lysine g/Mcal ME	2.81	3.16
Lysine-to-Methionine ratio	2.68	2.70
**METHIONINE (MET) (n = 26)**		
RP Methionine (g)	20	-
RP Lysine (g)	0	-
Methionine g/Mcal ME	1.06	1.17
Lysine g/Mcal ME	2.64	3.16
Lysine-to-Methionine ratio	2.49	2.70
**LYSINE (LYS) (n = 26)**		
RP Methionine (g)	0	-
RP Lysine (g)	20	-
Methionine g/Mcal ME	0.85	1.17
Lysine g/Mcal ME	2.82	3.16
Lysine-to-Methionine ratio	3.31	2.70

^a^ Cornell Net Carbohydrate and Protein System version 6.55 [5].

**Table 5 animals-13-03118-t005:** Least squared means for average body weight (BW), milk yield (kg/d), milk fat (kg/d and %), milk protein (kg/d and %), and milk urea (mg/L) during 70 DIM and *p*-values for general linear mixed models for repeated measures effects.

Item	CON	LYS	MET	LYS × MET	SEM ^1^	*p*-Value for Effects						
						LYS	MET	LYS × MET	DAY	LYS × DAY	MET × DAY	LYS × MET × DAY
IBW, kg ^2^	667.5	665.4	669.8	678.3	18.53	-	-	-	-	-	-	-
FBW, kg ^3^	638.4	633.4	642.9	652.8	17.74	0.32	0.45	0.14	0.02	0.38	0.18	0.17
Milk, kg	34.7	34.9	36.8	36.5	2.74	0.63	0.10	0.14	0.001	0.56	0.29	0.64
Solids, %	12.8	12.7	12.9	12.9	0.32	0.85	0.65	0.18	0.16	0.38	0.45	0.43
Fat, kg	1.44 ^a^	1.63 ^b^	1.61 ^b^	1.69 ^b^	0.058	0.01	0.03	0.03	0.17	0.38	0.41	0.003
Fat, %	4.63 ^ab^	4.49 ^b^	4.52 ^b^	4.68 ^a^	0.076	0.41	0.18	0.02	0.001	0.88	0.76	0.54
Protein, kg	1.14 ^a^	1.23 ^ab^	1.25 ^b^	1.27 ^b^	0.039	0.06	0.01	0.02	0.65	0.92	0.01	0.05
Protein, %	3.45 ^a^	3.56 ^b^	3.47 ^a^	3.53 ^ab^	0.059	0.01	0.63	0.14	0.001	0.04	0.45	0.88
Urea (mg/L)	289.3	303.4	306.7	301.8	12.35	0.27	0.28	0.23	0.91	0.14	0.13	0.15

CON: Control, n = 26, not supplemented; LYS: lysine group, n = 26, 20 g/cow per day of RP lysine; MET: methionine group, n = 26, 20 g/cow per day of RP methionine; LYS + MET: lysine plus methionine group, n = 26, 20 g/cow per day of RP methionine plus 20 g/cow/day of RP lysine. ^1^ Standard Error of Mean, ^2^ Initial body weight, ^3^ Final body weight. Different letters within row statistical differences at p ≤ 0.05

## Data Availability

Data is available upon request.

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
