# Peer review of "An Evaluation of the Supplementation of Rumen-Protected Lysine and Methionine on the Lactation Performance of Fall Parturition Grazing Holstein Cows in Southern Chile"

_animals, 2023, doi:10.3390/ani13193118_

Round 1

Reviewer 1 Report

The manuscript ID animals-2547634 describes the effect of rumen-protected lysine and methionine on lactation performance in grazing dairy cows. This topic is research related to amino acid nutrition in dairy cows, focusing on grazing dairy cows in Chile. In addition to previously published information, this study adds information on the effects of rumen-protective lysine 20g and methionine 20g supplementation in autumn-calving dairy cows under grazing conditions in southern Chile. The materials and methods for this experiment are fine. The conclusions are consistent with the evidence and arguments presented. References are good.

I recommend that you revise your manuscript referring to the following.

L24, 37, 465 milk solids => milk fat and protein

L50-51 Associated with L418-422. Do you think this version 7 information is necessary? Are you thinking of comparing versions 6.5 and 7?

Table1 Why did you calculate at 75 days in milk?

L284 % there => percentage, there

L294 percentage there => percentage, there

L317 BW (kg) there => BW (ka), there

L418-422, L431-434 Associated with L50-51. If you compare versions 6.5 and 7, add the data to the table to show the difference.

Author Response

Reviewer 1

The manuscript ID animals-2547634 describes the effect of rumen-protected lysine and methionine on lactation performance in grazing dairy cows. This topic is research related to amino acid nutrition in dairy cows, focusing on grazing dairy cows in Chile. In addition to previously published information, this study adds information on the effects of rumen-protective lysine 20g and methionine 20g supplementation in autumn-calving dairy cows under grazing conditions in southern Chile. The materials and methods for this experiment are fine. The conclusions are consistent with the evidence and arguments presented. References are good.

R: Thank you very much for your insights and recommendations.

I recommend that you revise your manuscript referring to the following.

 L24, 37, 465 milk solids => milk fat and protein

R: Corrected as indicated.

L50-51 Associated with L418-422. Do you think this version 7 information is necessary? Are you thinking of comparing versions 6.5 and 7?

R: Actually, we are not comparing versions. We are just giving information for the upcoming version that can help us to better explain the milk urea content of the present investigation. Definitely, the upcoming version will be more precise in protein and amino acid nutrition than the current one.

Table1 Why did you calculate at 75 days in milk?

R: This is because the current nutritional management of the farm considers a diet for milk yield at 75 days as the average days in milk for the high producing group. We wanted to follow the same approach taken by the farm nutritionist.  

L284 % there => percentage, there

R: Corrected as indicated.

L294 percentage there => percentage, there

R: Corrected. Semicolon added.

L317 BW (kg) there => BW (ka), there

R: Corrected as indicated.

L418-422, L431-434 Associated with L50-51. If you compare versions 6.5 and 7, add the data to the table to show the difference.

R: As we pointed out in the discussion, version 7.0 is not available for ration formulation yet, therefore we cannot add the suggested information. Version 7.0 has been not presented in detail. Only a general description of the version 7.0 has been shared by Cornell researchers in recent congresses

Reviewer 2 Report

Simple summary

L15-25 This section is a true copy of the abstract. It must be understood that the objective of the simple summary is to describe in a very brief and simple language the general context of the manuscript, including the problem, main findings and conclusion. Remember that this section applies to less or non-specialized readers-

Abstract

L17 It is not clear if only 26 cows were assigned to treatments, or 26 cows were assigned by treatment combination. In addition, the specific experimental design should be indicated, everything appears to be a 2 (with and without Met) x 2 (with and without Lys) factorial arrangement under a completely randomized design.

L31-36 Description of results should be rephrased. First, interaction effects could be described, and if they are not significant, then results by main factors could be described.

L36-38 Conclusion should be an overall idea of most outstanding findings, and consequently justify the supplementation or not these amino acids.

Introduction

L95 Add citation of the study referred.

L96 Add hypothesis before objective.

Materials and Methods

Add the letter number granted by the Institutional Ethics Committee. This is important because it guarantees that the procedures and management carried out with the animals were made in accordance with the legality of the country's animal laws.

L152 Describe the experimental design in a correct way. 22 factorial arrangements under a completely random design.

L203-204 There is poor wording in this sentence so it is suggested to adjust it.

L250-251 Statistical analysis is incorrect because the model is not considering the fixed effect of lysine and methionine. In fact, the model was applied considering only the Lys x Met interaction, which is represented by the fixed effect of treatment group (Gi). It is not possible to assume that two factors interact and put it in the model as an effect of treatments. In all cases, the authors did not respect the experimental design used to assign the experimental units to the treatment combinations.

Therefore, all the information must be reanalyzed, including in the model the fixed effects of lysine, methionine, time, all the interactions between the three main factors, and finally the covariates and random effects.

Results, discussion and conclusions

Since the statistical analysis is incorrect, all the information described in these sections does not make sense to review it. So, after re-analyzing the data, the results section will very possibly change, and consequently, the discussion and conclusion. Therefore, I reserved the process of reviewing these sections for the next submission of the manuscript.

No

Author Response

Reviewer 2

Simple summary

L15-25 This section is a true copy of the abstract. It must be understood that the objective of the simple summary is to describe in a very brief and simple language the general context of the manuscript, including the problem, main findings and conclusion. Remember that this section applies to less or non-specialized readers-

 R: Corrected as indicated.

Abstract

L17 It is not clear if only 26 cows were assigned to treatments, or 26 cows were assigned by treatment combination. In addition, the specific experimental design should be indicated, everything appears to be a 2 (with and without Met) x 2 (with and without Lys) factorial arrangement under a completely randomized design.

R: Thanks for your comments. It is 26 cows per group. We corrected the abstract now. Regarding the study design, please see our detailed comments in the following responses.

L31-36 Description of results should be rephrased. First, interaction effects could be described, and if they are not significant, then results by main factors could be described.

R: This paragraph has been modified following the reviewer’s comments.

L36-38 Conclusion should be an overall idea of most outstanding findings, and consequently justify the supplementation or not these amino acids.

R: Modified as suggested.

 Introduction

L95 Add citation of the study referred.

R: Added.

L96 Add hypothesis before objective.

 R: Added.

Materials and Methods

Add the letter number granted by the Institutional Ethics Committee. This is important because it guarantees that the procedures and management carried out with the animals were made in accordance with the legality of the country's animal laws.

R: Unfortunately, Chile has not an overall official ethical committee to conduct research, but dairy producers are voluntarily accredited for animal wellbeing high standards at the Chilean Dairy Consortium. The owner of the farm now has submitted a letter of consent to the journal office and editors to endorse and authorize this study under the Chilean Dairy Consortium requirements.

L152 Describe the experimental design in a correct way. 22 factorial arrangements under a completely random design

R: The original objective of this study was to compare 4 independent groups with different Lysine to Methionine ratio. As a result, we were expecting better responses for the group supplemented with both amino acids (Lys to Met = 2.68; optimal = 2.70). This was added as the hypothesis in the text now. We admit that In our first submission we committed a mistake by claiming a 2 by 2 factorial randomized experimental design. Our intention was always to compare the 4 independent groups for outcomes over time and compare the parallelism of the curves over time (repeated measures analysis).

For this purpose, the MIXED procedure of SAS 9.4 was utilized (SAS/STAT, SAS Inst. Inc., Cary, NC, U.S.A). This procedure allows for testing random effects in the model and permits modeling the covariance structure of the data set. The advantage of modeling the covariance structure is particularly central for repeated measures, since measurements obtained closer in time are potentially more correlated than those obtained through more distant times. Consequently, this approach is more consistent and produce more accurate estimates of variances and correlations. We use as a reference and base knowledge the paper by Littell et al., 1998 (Littell RC, Henry PR, Ammerman CB. Statistical analysis of repeated measures data using SAS procedures. J Anim Sci. 1998;76(4):1216-31. doi: 10.2527/1998.7641216x), as he is one of the developers of the mixed model approach for repeated measures in SAS (SAS for Mixed Models, Second Edition, Ramon C. Littell, Ph.D., George A. Milliken, Ph.D., Walter W. Stroup, Ph.D., Russell D. Wolfinger, Ph.D., Oliver Schabenberger, Ph.D. SAS Institute, 25 Jun 2007 - Mathematics - 828 pages). In this case, the main effect of treatment group (Control, LYS, MET, LYS+MET) is a fixed effect, the effect of time (week) is also a fixed effect, consequently the effect of the interaction group by time is also a fixed effect. However, the random effect is the cow (experimental unit) nested within treatment group.

Having said this, the treatment by day interaction is the most important effect of the model because it determined the parallelism of the curves among treatments. Schwarz’s Bayesian Criterion was used to test the goodness of fit of each particular model based on the best covariance structure. Since the error term may increase while multiple comparisons are performed simultaneously, a corrected multiple comparison test is necessary to control the error rate to an appropriate level. Then, the Tukey-Kramer test was used as a post-hoc testing to determine whether there is a difference between the mean of all possible pairs using a studentized range distribution based on Lee S, Lee DK. What is the proper way to apply the multiple comparison test? Korean J Anesthesiol. 2018;71(5):353-60. Epub 20180828. doi: 10.4097/kja.d.18.00242.

Finally, but not less important, and being very humble, as the first author of this manuscript, I feel honored and privileged that Dr. Ramon Littell was member of my Master and PhD committee when I fulfilled my graduate studies at the University of Florida. I had the opportunity to publish 6 papers with him (see below), who provided me with the advice of all my statistical analysis which are similar to the current study we are intending to publish here in Animals (Basel).  

1: Melendez P, Goff JP, Risco CA, Archbald LF, Littell R, Donovan GA. Pre-partum

monensin supplementation improves body reserves at calving and milk yield in

Holstein cows dried-off with low body condition score. Res Vet Sci. 2007

Jun;82(3):349-57. doi: 10.1016/j.rvsc.2006.09.009.

2: Melendez P, Goff JP, Risco CA, Archbald LF, Littell RC, Donovan GA. Effect of

administration of a controlled-release monensin capsule on incidence of calving-

related disorders, fertility, and milk yield in dairy cows. Am J Vet Res. 2006

Mar;67(3):537-43. doi: 10.2460/ajvr.67.3.537.

3: Melendez P, Goff JP, Risco CA, Archbald LF, Littell R, Donovan GA. Incidence

of subclinical ketosis in cows supplemented with a monensin controlled-release

capsule in Holstein cattle, Florida, USA. Prev Vet Med. 2006 Jan 16;73(1):33-42.

doi: 10.1016/j.prevetmed.2005.08.022.

4: Melendez P, Goff JP, Risco CA, Archbald LF, Littell R, Donovan GA. Effect of

a monensin controlled-release capsule on rumen and blood metabolites in Florida

Holstein transition cows. J Dairy Sci. 2004 Dec;87(12):4182-9. doi:

10.3168/jds.S0022-0302(04)73562-6.

5: Melendez P, Donovan GA, Risco CA, Littell R, Goff JP. Effect of calcium-

energy supplements on calving-related disorders, fertility and milk yield during

the transition period in cows fed anionic diets. Theriogenology. 2003 Sep

15;60(5):843-54. doi: 10.1016/s0093-691x(03)00103-1.

6: Melendez P, Donovan A, Risco CA, Hall MB, Littell R, Goff J. Metabolic

responses of transition Holstein cows fed anionic salts and supplemented at

calving with calcium and energy. J Dairy Sci. 2002 May;85(5):1085-92. doi:

10.3168/jds.S0022-0302(02)74169-6.

L203-204 There is poor wording in this sentence so it is suggested to adjust it.

R: Modified as suggested.

L250-251 Statistical analysis is incorrect because the model is not considering the fixed effect of lysine and methionine. In fact, the model was applied considering only the Lys x Met interaction, which is represented by the fixed effect of treatment group (Gi). It is not possible to assume that two factors interact and put it in the model as an effect of treatments. In all cases, the authors did not respect the experimental design used to assign the experimental units to the treatment combinations.

Therefore, all the information must be reanalyzed, including in the model the fixed effects of lysine, methionine, time, all the interactions between the three main factors, and finally the covariates and random effects.

R: We hope the above explanation may help clarify our statistical approach. We were not interested to test an interaction effect of Lysine x Methionine. Our objective was to compare 4 independent groups based on the Lys to Met ratio, which the Lys plus Met group had the best ratio.

 Results, discussion and conclusions

Since the statistical analysis is incorrect, all the information described in these sections does not make sense to review it. So, after re-analyzing the data, the results section will very possibly change, and consequently, the discussion and conclusion. Therefore, I reserved the process of reviewing these sections for the next submission of the manuscript.

R: We respectfully hope that the reviewer will consider our arguments for the experimental design and statistical approach.

Reviewer 3 Report

Comments and suggestions for Authors

The „Simple Summary” needs rewriting. This paragraph should be written for a lay audience, i.e., no abbreviations and such detailed of methodology and result. In this form it is basically no different from “Abstract”.

I suggest completing the paragraph “Introduction”. The introduction should briefly place the study in a broad context and the current state of the research field should be reviewed carefully and key publications cited. In my opinion some information should be rather in the section “Discussion”.

All the figures should contain a legend (information about groups). Tables and figures should contain descriptions accurate enough to allow analysis without knowledge of the text of the work.

All items in the section “References” need improvement in terms of the requirements of the Animals journal (punctuation marks, bold, italics) – guide for authors.

Minor remarks

Ln 28-29: In the group LYS+MET both amino acids were in the rumen protected form. Please clarify this information in the text.

Ln 125: The abbreviation should be explained when first used (Dietary Cation Anion Difference).

Ln 132: The abbreviation should be explained.

Ln 183: Neutral detergent fiber.

Ln 188: Fresh pasture, Pasture silage, Barley straw.

Ln 188: Crude protein, Solube protein, Solue fiber.

Ln 189: Explain in a legend abbreviation aNDFom.

Ln 283: Milk fat.

Ln 236: a Cornell Net Carbohydrate and Protein System version 6.55 [5].

Ln 327: Text in table 5 – Milk yield, Total solids (font size as in the text of the work).

Ln 328-329: n=26 – spaces

Ln 329: In group LYS+MET, the proportion of RP lysine was 20 g/cow per day.

Ln 330: 2 Final body weight.

Ln 331-332: I sugest „Milk yield (kg/day) during the first 70 days postpartum in fall grazing Holstein cows” insted of  „Milk yield (kg/day) for CON, LYS, MET, and LYS+MET during the first 70 days postpartum in multiparous grazing Holstein cows in the south of Chile”. This remark also applies to other figures.

Ln 335 and 339: Milk fat (…).

Ln 345 and 349: Milk protein. * P = 0.04 Con vs Met

Author Response

Reviewer 3

The „Simple Summary” needs rewriting. This paragraph should be written for a lay audience, i.e., no abbreviations and such detailed of methodology and result. In this form it is basically no different from “Abstract”.

R: Thanks for your comments. The simple summary was re-written and simplified.

I suggest completing the paragraph “Introduction”. The introduction should briefly place the study in a broad context and the current state of the research field should be reviewed carefully and key publications cited. In my opinion some information should be rather in the section “Discussion”.

R: We agree. We considered your suggestion in the revised document now.

All the figures should contain a legend (information about groups). Tables and figures should contain descriptions accurate enough to allow analysis without knowledge of the text of the work.

R: Modified as indicated.

All items in the section “References” need improvement in terms of the requirements of the Animals journal (punctuation marks, bold, italics) – guide for authors.

R: Thanks. We used the endnote software to arrange the references based on the Animals (Basel) format. We have re-checked each reference format now

Minor remarks

Ln 28-29: In the group LYS+MET both amino acids were in the rumen protected form. Please clarify this information in the text.

R: Modified as indicated.

Ln 125: The abbreviation should be explained when first used (Dietary Cation Anion Difference).

R: Modified as indicated.

Ln 132: The abbreviation should be explained.

R: Modified as indicated.

Ln 183: Neutral detergent fiber.

R: Corrected.

Ln 188: Fresh pasture, Pasture silage, Barley straw.

R: corrected

Ln 188: Crude protein, Solube protein, Solue fiber.

R: corrected

Ln 189: Explain in a legend abbreviation aNDFom.

R: Modified as indicated.

Ln 283: Milk fat.

R: Modified as indicated.

Ln 236: a Cornell Net Carbohydrate and Protein System version 6.55 [5].

R: Modified as indicated.

Ln 327: Text in table 5 – Milk yield, Total solids (font size as in the text of the work).

R: Modified as indicated.

Ln 328-329: n=26 – spaces

R: Modified as indicated.

Ln 329: In group LYS+MET, the proportion of RP lysine was 20 g/cow per day.

R: Modified as indicated.

Ln 330: 2 Final body weight.

R: Modified as indicated.

Ln 331-332: I sugest „Milk yield (kg/day) during the first 70 days postpartum in fall grazing Holstein cows” insted of  „Milk yield (kg/day) for CON, LYS, MET, and LYS+MET during the first 70 days postpartum in multiparous grazing Holstein cows in the south of Chile”. This remark also applies to other figures.

R: Modified as indicated.

Ln 335 and 339: Milk fat (…).

R: Modified as indicated.

Ln 345 and 349: Milk protein. * P = 0.04 Con vs Met

R: Modified as indicated.